# Therapeutic Use of Valproic Acid and All-Trans Retinoic Acid in Acute Myeloid Leukemia—Literature Review and Discussion of Possible Use in Relapse after Allogeneic Stem Cell Transplantation

**DOI:** 10.3390/ph14050423

**Published:** 2021-05-02

**Authors:** Øystein Bruserud, Galina Tsykunova, Maria Hernandez-Valladares, Hakon Reikvam, Tor Henrik Anderson Tvedt

**Affiliations:** 1Department of Clinical Science, University of Bergen, N-5021 Bergen, Norway; hakon.reikvam@uib.no; 2Department of Medicine, Haukeland University Hospital, N-5021 Bergen, Norway; galina.tsykunova@helse-bergen.no (G.T.); tor.henrik.anderson.tvedt@helse-bergen.no (T.H.A.T.); 3The Proteomics Facility of the University of Bergen (PROBE), University of Bergen, N-5021 Bergen, Norway; maria.hernandez-valladares@uib.no

**Keywords:** acute myeloid leukemia, valproic acid, all-trans retinoic acid, allogeneic stem cell transplantation, relapse

## Abstract

Even though allogeneic stem cell transplantation is the most intensive treatment for acute myeloid leukemia (AML), chemo-resistant leukemia relapse is still one of the most common causes of death for these patients, as is transplant-related mortality, i.e., graft versus host disease, infections, and organ damage. These relapse patients are not always candidates for additional intensive therapy or re-transplantation, and many of them have decreased quality of life and shortened expected survival. The efficiency of azacitidine for treatment of posttransplant AML relapse has been documented in several clinical trials. Valproic acid is an antiepileptic fatty acid that exerts antileukemic activity through histone deacetylase inhibition. The combination of valproic acid and all-trans retinoic acid (ATRA) is well tolerated even by unfit or elderly AML patients, and low-toxicity chemotherapy (e.g., azacitidine) can be added to this combination. The triple combination of azacitidine, valproic acid, and ATRA may therefore represent a low-intensity and low-toxicity alternative for these patients. In the present review, we review and discuss the general experience with valproic acid/ATRA in AML therapy and we discuss its possible use in low-intensity/toxicity treatment of post-allotransplant AML relapse. Our discussion is further illustrated by four case reports where combined treatments with sequential azacitidine/hydroxyurea, valproic acid, and ATRA were used.

## 1. Introduction

Acute myeloid leukemia (AML) is an aggressive malignancy that can only be cured by intensive chemotherapy possibly followed by autologous or allogeneic stem cell transplantation [1,2]. The median age at the time of diagnosis is 65–70 years, and the large group of elderly patients above 70–75 years of age as well as younger unfit patients cannot receive the most intensive treatment; instead, these patients usually receive various forms of leukemia-stabilizing therapy (e.g., hypomethylating treatment) or only palliative care [1].

Relapse is a common cause of death after potentially curative intensive conventional chemotherapy, and a relatively large group of patients relapse even after the most intensive treatment including allogeneic stem cell transplantation [1,2]. There are several therapeutic alternatives for patients with AML relapse and for allotransplant recipients, including conventional intensive chemotherapy possibly followed by transplantation/re-transplantation after remission induction or less intensive treatment to achieve disease stabilization [1,2]. However, many of these patients are unfit for further intensive therapy and have reduced quality of life, and low-toxicity treatments are probably the best alternative for such patients. Low-toxicity therapy may also be the best alternative for several patients with early relapse and high AML cell burden because these patients often have aggressive chemo-resistant disease. Finally, the same low-toxicity alternatives should also be considered for other elderly or unfit patients with newly diagnosed AML, including patients with high-risk AML [3].

Several experimental and clinical studies have investigated the effects of ATRA combined with valproic acid in the treatment of AML. Important characteristics of the two drugs are summarized in Table 1 [3,4,5,6,7,8,9,10,11,12,13,14,15,16,17,18]. Additional studies are needed to further characterize the possible mechanisms for interactions between the two drugs at the cellular level, especially whether such interactions differ between patients and depend on karyotype and/or molecular genetic abnormalities.

Valproic acid is a fatty acid that is used as an antiepileptic drug; it also has anti-AML effects probably due to its function as a histone deacetylase inhibitor and a systemic metabolic modulator (Table 1) [3,5]. Its side effects are well characterized, and severe toxicity is uncommon, dose-dependent, and reversible [3]. All-trans retinoic acid (ATRA) is widely used in the treatment of acute promyelocytic leukemia (APL), but it may also have antileukemic effects at least for certain patients with non-APL variants of AML (Table 1) [4]. Valproic acid has therefore been combined with ATRA in several clinical studies of unfit or elderly patients with non-APL AML [3]. Further investigation of this dual drug combination is also encouraged by the authors of a recent randomized clinical study who combined hypomethylating therapy with valproic acid and ATRA [19]. In this article, we review the scientific basis for and discuss whether ATRA/valproic acid should be further investigated as a possible part of leukemia-stabilizing treatments for elderly or unfit patients with non-APL variants of AML, including patients with leukemia relapse after allogeneic stem cell transplantation.

## 2. Experimental and Clinical Studies of Valproic Acid in Non-APL Variants of AML

### 2.1. Experimental Studies of Valproic Acid Effects on AML Cells

The antileukemic effects of valproic acid wer investigated in several previous studies. Studies in a murine xenograft model based on the Kasumi-1 AML cell line with the *RUNX1-RUNX1T1* fusion oncogene (i.e., t(8;21) translocation) suggest that valproic acid upregulates histone acetylation in the *p21* promoter region, leading to increased p21 expression, suppressed phosphorylation of the retinoblastoma protein, and finally induction of G0/G1 arrest [20]. This is possibly due to histone deacetylase (HDAC) 1 inhibition with relocation of this enzyme from the nuclear to the perinuclear region [21]. At the same time, histone H3 and H4 hyperacetylation causes transcriptional reactivation of genes that have been silenced by the fusion protein. The p21 upregulation caused by in vivo valproic acid therapy was also observed in primary AML cells without *RUNX1-RUNX1T1* fusion [22]. However, additional mechanisms may contribute to the antiproliferative and proapoptotic effects of valproic acid. First, the hippo signaling pathway is important for regulation of proliferation and apoptosis, and altered signaling through this pathway (possibly due to the induction of *RASSF1A* expression) may be an antileukemic mechanism especially in patients with normal and adverse karyotype [23]. Second, valproic acid in combination with interferon (IFN)-α alters the activation/phosphorylation of Akt, ERK1/2, p38, and p53 [24]. Third, a small study suggested that the CXCR4 protein level is reduced by valproic acid at least for certain patients and especially for AML cells that do not express the CD34 stem cell marker [25]. However, valproic acid may also increase the expression of genes involved in chemoresistance, e.g., MAKKPK2, HSP90AA1, HSP90AB1, and ACTB [24], and such effects may then counteract the other antileukemic effects.

The studies described above suggest that the antileukemic effects of valproic acid are mediated through several molecular mechanisms that possibly differ between patients. A previous in vitro study investigated patient heterogeneity with regard to the functional effects of valproic acid and three other HDAC inhibitors on primary AML cells [26]. Two main patient subsets were then identified; one subset showed a dose-dependent antiproliferative effect of various inhibitors, whereas the other subset showed growth enhancement when testing intermediate concentrations and inhibitory effects only when testing higher concentrations. These dose-dependent differences were observed in the presence of both ATRA and the hypomethylating agent decitabine, and during co-culture with AML-supporting bone marrow stromal cells. The two patient subsets differed in AML cell expression of several genes involved in the regulation of cell cycle progression, gene transcription, and DNA damage responses. In contrast, normal CD34^+^ hematopoietic cells showed only growth enhancement for intermediate concentrations but no antiproliferative effect with higher concentrations. Thus, AML patients are heterogeneous with regard to the effects of valproic acid on their leukemia cells, and this heterogeneity may be caused by differences in the balance between genes that mediate chemosensitivity versus chemoresistance, as described above.

### 2.2. In Vivo Effects of Valproic Acid in AML: Clinical Studies and Animal Models

Several studies have investigated the effects of valproic acid on AML cells derived from patients receiving valproic acid treatment. First, valproic acid monotherapy with serum concentrations corresponding to the accepted therapeutic level can induce hyperacetylation of histones H3 and H4, signs of myelomonocytic differentiation, and hematological improvement according to the response criteria for myelodysplastic syndromes (MDS) [27]. These responses can last for 60–180 days, at least for a subset of patients [27]. Second, another study investigated the in vivo effects of combined ATRA, valproic acid, and theophylline treatment, and pretreatment AML cells for responders and nonresponders to this triple combination differed especially in their expression of genes involved in transcriptional regulation [28]. The initial two days of ATRA monotherapy in this study decreased the expression of several HOX genes, whereas the triple combination altered the mRNA expression of DNA methyl transferase 3A (*DNMT3A*) (i.e., an epigenetic regulator) together with several genes involved in G-protein coupled receptor signaling, and the combination decreased the protein levels of Gata-2, NFκB p65, and Bcl-2 in AML cells [22]. Third, Rücker et al. [29] compared the gene expression profiles for AML cells derived from patients receiving intensive chemotherapy with and without additional valproic acid, and valproic acid then altered the expression of genes important for cell cycle regulation, DNA repair, and apoptosis. However, all of these studies are relatively small and the results should therefore be interpreted with great care, but they are supported by two studies in animal models. Valproic acid induced G0/G1 arrest in the murine xenograft model of the Kasumi-1 cell line [30], and valproic acid monotherapy increased the survival of immune-competent Brown Norwegian myeloid leukemia rats [31].

The effect of in vivo therapy based on ATRA plus valproic acid on the proteomic and phosphoproteomic profiles of human AML cells was investigated in a recent study [32]. This in vivo study showed that both ATRA and valproic acid had complex biological effects on the AML blasts involving several fundamental cellular functions; these observations are discussed in Section 3.6.

### 2.3. Effects of Valproic Acid on Normal Hematopoietic and Bone Marrow STROMAL cells

Experimental studies of human and murine hematopoietic stem cells show that valproic acid increases their proliferation and self-renewal capacity [11]. The drug accelerates cell cycle progression and decreases p21 levels, and these effects are possibly caused by the inhibition of GSK3β, altered Wnt signaling, and the stabilization of β-catenin. The final effect seems to be an increased expression of HoxB4 leading to an altered regulation of stem cell self-renewal. Both β-catenin and HoxB4 can increase normal stem cell proliferation and expand the stem cell pool, and these effects may contribute to the increased levels of normal peripheral blood cell counts seen during valproic acid therapy. Valproic acid can also decrease VLA-4 expression in hematopoietic cells [33] and seems to have an antiproliferative effect on osteoblastic cells [34]; these effects together with the effects on endothelial cells (see the next section) may alter the AML supportive functions of these stromal cells in bone marrow stem cell niches.

The effect of valproic acid on AML-associated angiogenesis has been investigated in a murine xenograft model using the Kasumi-1 AML cell line [20]. Valproic acid then had an antileukemic effect with reduced expressions of proangiogenic VEGF, bFGF, and the VEGFR2 receptor, thereby leading to reduced microvessel density. The observed reduction in serum levels of the endothelium-associated proteoglycan endocan in patients receiving valproic acid may also reflect direct effects on endothelial cells with altered angio-regulation [35], and this is further supported by a study on microvascular endothelial cells describing decreased proliferation and migration/tube formation together with an altered cytokine release profile by endothelial cells after exposure to valproic acid [34].

### 2.4. Effects of Valproic Acid on Normal Immunocompetent Cells

Valproic acid affects several immunocompetent cells, and immunosuppressive effects seem to be most common. The effects for various cell types are described more in detail below.
T cells. Valproic acid has strong antiproliferative but not proapoptotic effects on T cells in the presence of cytarabine, and it can also alter the expression of the activation markers CD38 and CD69 as well as the release of FasL, heat shock protein (HSP)90, and various cytokines [36]. The levels of regulatory T cells are increased during AML stabilizing treatment with ATRA and valproic acid [37].Dendritic cells. Valproic acid inhibits the release of IFN-α, TNFα, and IL-6 by plasmacytoid dendritic cells; reduces their expression of costimulatory molecules as well as their capacity to promote CD4^+^ T cell proliferation and IFN-γ production; and increases the proportion of anti-inflammatory IL-10-positive T cells [38]. The drug also downregulates group I CD1 expression; reduces the secretions of IL-6, IL-10, and IL-23 and TNF-α release; increases IL-8 release; and reduces the capacity to promote differentiation of Th17 cells in human monocyte-derived immature dendritic cells [39]. Furthermore, a third study showed that valproic acid causes downregulation of the (co)stimulatory molecules CD40, CD80, CD83, CD86, and HLA-DR and decreases IL-10 and IL-12p70 production in mature dendritic cells [40]. These last authors also suggested that the proportion of IFN-γ^+^CD4^+^ alloreactive T cells and the granzyme B expression by CD8^+^ T cells were both decreased when these cells were cocultured with dendritic cells that were previously exposed to valproic acid. Finally, HDAC inhibition can reduce CD1a expression together with the capacity of chemokine-induced migration, immunostimulatory capacity, and cytokine release in monocyte-derived dendritic cells [41]. These last effects seem to be caused by inhibited signaling through NFκB, IRF-3, and IRF-8.Monocytes/macrophages. Low levels of valproic acid alter the differentiation of human monocyte-derived macrophages and their expressions of CXCL8/IL-8, IL-1β, IL-6, TNFα, and IL-10 (but not IL-12), whereas higher valproic acid levels reduce the release of cytokines in general [42]. Furthermore, valproic acid can enhance the expansion of nonclassical macrophages [43], with an overexpression of CD163 (but no effect on CD86 expression), decreased IL1-β and TNFα expression, as well as increased expressions of anti-inflammatory IL-10 and TGF-β1.Mesenchymal stem/stromal cells (MSCs). Valproic acid increases the glycolytic, respiratory, and T-cell suppressive capacity of MSCs [44]; increases their CXCR7 expression; increases the migratory capacity of bone marrow MSCs [45]; and induces osteogenic differentiation of human MSCs [33].

To conclude, valproic acid can modulate the function of several immunocompetent/immunomodulatory cells, but it is difficult to know whether this immunomodulation is important for direct (e.g., cytotoxic activity) or indirect (e.g., regulation of angiogenesis through cytokine or protease release) antileukemic effects of various immunocompetent cell subsets in vivo.

### 2.5. Altered Systemic Mediator Levels and Metabolic Regulation during Treatment with Valproic Acid in AML Patients

Valproic acid alters the serum levels of many different soluble mediators [46], including interleukins, chemokines, and growth factors for hematopoietic as well as endothelial cells. The levels of soluble HSPs, inducing HSP70 and HSP90 that may be associated with prognosis in human AML [46], as well as the levels of soluble endocan (i.e., a proteoglycan) are altered [35].

Valproic acid therapy alters the systemic serum metabolic profiles of AML patients [5]: it had minor effects on glucose metabolism but caused extensive alteration of lipid and amino acid metabolite levels. These effects on systemic levels of soluble mediators may influence immunocompetent cells and/or the functional status and therefore the chemosensitivity of AML cells through modulation of intracellular signaling (e.g., ligation of Toll-like receptors by metabolites or cytokines) or through altered epigenetic regulation by, for example, lipid/fatty acid metabolites [12,47,48,49,50,51,52,53].

## 3. Experimental and Clinical Studies of ATRA in Non-APL Variants of AML

### 3.1. Experimental and Clinical Studies of ATRA Effects on AML Cells

The effects of ATRA and other vitamin A derivatives in non-APL AML have been reviewed in several recent excellent reviews [4,6,54,55,56,57], and for this reason, these effects are briefly described.

Experimental studies suggest that the antileukemic effect of ATRA varies between patients [4,58], and this is further supported by clinical studies. A large phase III clinical trial including 242 elderly patients above 60 years of age described higher response rates and better event-free as well as overall survival in patients receiving ATRA together with conventional intensive induction and first consolidation chemotherapy [59]. A later reanalysis of this study suggested that the favorable effect was restricted to the subset of patients with *NPM1* mutations with *FLT3*-wild type (wt) [60]. Another smaller study also including elderly patients described a similar improvement when patients received ATRA in addition to standard intensive chemotherapy, although this beneficial effect was observed only for patients with low levels of the transcriptional cofactor MNl [61]. However, other studies have described no effect of ATRA when combined with conventional intensive chemotherapy, and this was true both for analyses of the overall patient populations as well as patient subset analyses [62,63]. Furthermore, a recent meta-analysis including eight previous studies also concluded that ATRA did not have any effect in non-APL variants of AML when combined with conventional intensive treatment [64]. It should be emphasized that these previous studies of ATRA differed with regard to several parameters, e.g., patient age, cytotoxic drug treatment, and treatment schedules. Finally, two recent studies suggested that ATRA may have an effect when combined with hypomethylating agents (see Section 2.2) [19,65], and especially azacitidine seems to be effective at least in the treatment of AML relapse after allotransplantation (see Section 7.1).

Several studies suggest that the effects of ATRA on primary non-APL AML cells vary between patients. The following factors seem to be associated with susceptibility to ATRA:t(8;21) AML. The *RUNX1-RUNX1T1* translocation seems to confer ATRA resistance [66], although this has been questioned by another study [67].*FLT3*-ITD. Studies in an animal model of *FLT3*-ITD/NPM1c-driven AML suggest that ATRA has an antileukemic effect in this AML model through effects on leukemic stem cells, but this pharmacological effect is counteracted by experimental expression of the stem cell-associated transcription factor *EVI1* [58].*NPM1* mutations. Experimental studies suggest that ATRA can lead to degradation of mutated *NPM1* with re-localization of *NPM1*-wt encoded by the second gene and therefore can induce cell cycle arrest, differentiation, and increased sensitivity to conventional cytotoxic drugs [68,69]. These observations are supported by certain clinical studies, as described above [59,60].*IDH1* mutations. Experimental studies suggest that patients with *IDH1* mutations are more sensitive to ATRA [70].High *MN1* expression. *MN1* encodes a transcription cofactor that is a member of the RAR/RXR complex, and high expression is associated with resistance to ATRA [61].High *EVI1* expression. This gene encodes a stem cell-associated transcription factor that is downregulated during hematopoietic differentiation. High expression is observed for 10% of AML patients and is associated with an adverse prognosis [58,71,72,73]. This factor is upregulated by ATRA [74,75,76], and high EVI1 levels seem to enhance the transcriptional response to ATRA [77], thereby causing differentiation induction, decreased clonogenic proliferation, and decreased AML engraftment in immunocompromised mice [78]. These antileukemic effects seem to be caused by the effects of ATRA on the AML stem cells for patients with stem cell-derived leukemia and therefore high *EVI1* expression, but the effect is weak or absent in AML derived from progenitor cell transformation and shows low *EVI1* expression [79]. However, this effect of EVF1 expression is probably more complex because it may be further modulated by other genetic abnormalities [58].

Additional pharmacological factors may also influence the effect of ATRA. First, bone marrow stromal cells can degrade ATRA and therefore limit the antileukemic effects [67]. Second, the aldehyde dehydrogenase (ALDH) enzymes are involved in the synthesis of retinoids and metabolism of reactive aldehydes [80,81]. ATRA can downregulate ALDH activity and therefore sensitize AML cells to cytotoxic drugs [80]. A recent proteomic study demonstrated increased ALDH activity especially in primary AML cells derived from elderly patients [82], an observation suggesting that the antileukemic effects of ATRA may also be associated with patient age. Taken together, these observations suggest that the antileukemic effects of ATRA differ between patients, and this may explain the discrepancies between clinical studies regarding the antileukemic effects of ATRA.

### 3.2. Effects of ATRA on Normal Hematopoietic and Bone Marrow Stromal Cells

ATRA has several effects on normal hematopoiesis and is important both for granulopoiesis and erythropoiesis. A detailed review of these effects is outside the scope of this article, but a detailed discussion with additional references can be found in recent reviews [83,84,85]. ATRA may also influence both normal and leukemic hematopoiesis indirectly through its effects on MSCs, osteoblasts, and endothelial cells in the bone marrow microenvironment [86,87,88,89,90].

### 3.3. Effects of Vitamin A/ATRA on Normal Immunocompetent Cells

The role of vitamin A in immunoregulation has been reviewed in several recent articles [85,91,92,93,94,95], and these effects are therefore only briefly summarized. Firstly, the importance of vitamin A and its derivatives for immunity is illustrated by vitamin A deficiency that leads to decreased humoral and cellular responses, inadequate immune regulation, weak responses to vaccines, and poor lymphoid organ development [92,94]. Vitamin A can alter the levels and/or the function of several T cell subsets, including both proinflammatory subsets as well as Treg cells [92,93,94,95]. With regard to ATRA, this agent can both facilitate the development of Treg cells [94,95,96] and enhance anticancer immune reactivity [97]. Second, retinoic acid can alter trafficking/homing of immunocompetent cells, including both T and B cells [93]. Third, retinoic acid/ATRA can modulate the function of other immunocompetent cells [92,93]. Vitamin A is an important regulator of several innate immune cells, e.g., causing altered function of monocytes/macrophages and Innate lymphoid cells together with reduced levels of myeloid-derived suppressor cells in patients receiving Car-T cell therapy [92,93,98]. The altered differentiation and trafficking of monocytes/macrophages by ATRA then results in increased differentiation towards the M2 phenotype with inhibition of macrophage-mediated immunity and favored development of tolerance [87,88,94,99]. Finally, the effect of retinoic acid on the balance between immunity versus tolerance is further modulated by the microenvironment. Thus, vitamin A and probably also ATRA have extensive effects on both the development and the function of several immunocompetent cell subsets, and the effect of ATRA treatment will probably be difficult to predict in AML patients both during conventional intensive chemotherapy and following allogeneic stem cell transplantation.

The conclusion from several previous reviews is that the effects of vitamin A on immunoregulation and immunocompetent cells will depend on the dose, the tissue, and the biological context of the immune response [85,93,94]. Retinoic acid seems to modulate Treg differentiation in steady state/homeostasis but promotes activation of proinflammatory cell subsets during ongoing immune responses [85,93,94]. Both auto- and allotransplant recipients will have a T cell defect lasting several months after the hematopoietic reconstitution with normalization of peripheral blood neutrophil, monocyte subset, and thrombocyte counts [100,101,102]. It is not known how ATRA/vitamin A deficiency will influence this developing posttransplant immune system. Furthermore, allotransplant recipients seem to have an altered vitamin A metabolism during the first 28 days posttransplant compared with the pretransplant vitamin A status; these alterations are probably due to conditioning/reconstitution/pancytopenia rather than the early posttransplant nutritional status [103]. Finally, local production of retinoids by immunocompetent cells (monocytes and dendritic cell) has important immunoregulatory functions, but it is not known how high systemic doses will influence local retinoid-induced immunoregulation in various microenvironments. These overall data further illustrate that the immunological effects of posttransplant ATRA therapy are very difficult to predict based on the currently available knowledge.

The effects of ATRA on immunocompetent cells will be of particular importance in allotransplant recipients because immune-mediated antileukemic activity contributes to the overall antileukemic effect of this strategy but especially because immune-mediated complications contribute to the posttransplant non-relapse mortality [104]. ATRA is important for trafficking of immunocompetent cells, including the homing of various immune cell subsets to the gut that is one of the organs commonly affected in both acute and chronic graft versus host disease (GVHD) [105,106]. The importance of vitamin A for immunoregulation in allotransplant recipients is suggested by several observations. First, studies in animal models have shown that vitamin A deficiency alters the phenotype of acute GVHD; posttransplant T cell homing to the gut is then reduced in vitamin A-deficient mice, but other organs show more serious affection [107]. Second, chronic vitamin A deficiency changes the composition of the T cell compartment of donor mice with a reduction in the percentage of CD4^+^ T cells, and a decreased proportion of donor CD4^+^ T cells in marrow grafts leads to reduced incidence and severity of GVHD [108]. This study also showed that treatment with a pan-RAR antagonist inhibited donor T cell RAR signaling, and reduced the T cell alloreactivity and its ability to cause lethal GVHD. Third, in another animal study, exogenous RA significantly increased the expression of gut-homing molecules (CCR9 and α4β7 integrin) on donor T cells and augmented the accumulation of proinflammatory CD4^+^ and CD8^+^ T cells in the gut mucosa, leading to an exacerbation of colonic GVHD with increased mortality [109]. Retinoic acid depletion in the recipient mice then reduced the expression of gut-homing molecules in donor T cells after HSCT and attenuated the ability of these cells to cause lethal GVHD. Finally, the function of both host and donor dendritic cells was also influenced by the vitamin A status and this influences the risk of GVHD in this murine model [110].

The results from animal studies are in contrast with the observations from two studies in humans. A study of 114 consecutive pediatric allotransplant recipients demonstrated low levels of vitamin A at day +30 posttransplant to be associated with increased gastrointestinal GVHD at day +100 (38% vs. 12.4%, *p* = 0.0008). The same was true for treatment-related mortality (17.7% vs. 7.4% at 1 year, *p* = 0.03), and bloodstream infections were increased in patients with low vitamin A levels (24% vs. 8% at 1 year, *p* = 0.03) [111]. Surprisingly, the expression of the gut homing receptor CCR9 on T-effector memory cells 30 days after transplant was increased in children with low vitamin A levels [111]. Another study suggested that low vitamin A levels are also associated with chronic GVHD [112].

Taken together, these observations suggest that ATRA therapy after allotransplantation influences the immunological functions of the stem cell recipients, but possible effects with regard to the development of GVHD are difficult to predict and further studies are needed.

### 3.4. Altered Metabolic Regulation during Treatment with ATRA in AML Patients

ATRA alters the systemic (i.e., serum) metabolic profiles in AML patients, especially the levels of several amino acid and lipid metabolites, but these effects are relatively weak compared with valproic acid [5]. ATRA has only minor effects on glucose metabolism [5].

### 3.5. The Complexity of Combining Valproic Acid and ATRA in the Treatment of AML

The results described previously in Section 2 and Section 3 show that valproic acid and ATRA have complex effects, including direct effects on the AML cells as well as indirect effects mediated by their influence on neighboring cells in the bone marrow microenvironment, e.g., normal hematopoietic cells, bone marrow stromal cells that support both normal and leukemic hematopoiesis, and various immunocompetent cells. Figure 1 summarizes the effects of the two pharmacological agents on various cell types and therefore illustrates the complexity of combined valproic acid/ATRA therapy.

### 3.6. Valproic Acid/ATRA Based Anti-AML Therapy: What Can We Learn from Proteomic and Phosphoproteomic Studies

A recent study investigated the in vivo effects of ATRA alone and ATRA/valproic acid-based therapy on the proteomic and phosphoproteomic profiles of primary human AML cells [32]. The results of these studies are summarized in Table 2. This study suggests that the pretreatment proteomic and phosphoproteomic profiles of the AML cells show significant differences. Furthermore, the analyses during ATRA/valproic acid in vivo therapy showed that the in vivo concentrations of both these drugs reached levels that could influence important cellular processes in AML cells, especially processes involving RNA processing/metabolism/splicing. However, additional functions were also affected. These observations also reflect the complexity of these pharmacological effects.

Even though both ATRA and valproic acid have effects on transcriptional regulation and gene expression, we regard their effects on protein levels and posttranscriptional protein modulation (i.e., phosphorylation) to be of particular importance because several new AML therapies target various proteins (e.g., bcl2 inhibitors) or protein modifications (e.g., receptor tyrosine kinase inhibitors). In our opinion, proteomic/phosphoproteomic studies are therefore an important part of the scientific basis for the design of future pharmacological combinations.

## 4. Clinical Studies of Valproic Acid and ATRA in AML

Several studies have investigated the effects of valproic acid monotherapy or combination therapy including ATRA or low-dose chemotherapy for elderly or unfit AML patients, including patients with high-risk or relapsed AML [3,22,35,37,113,114,115,116,117,118,119,120]. Although many of these studies are relatively small, some general conclusions can be made:Only a minority of patients (approximately 25–35%) respond to this treatment.Most responders show stabilized disease when using the MDS response criteria (see Section 6.2). Exceptional patients achieve complete hematological remission, but this is usually seen when valproic acid is combined with ATRA plus low-toxicity chemotherapy.The most common response is stabilized/increased peripheral blood platelet counts, and for most patients, the duration of such responses is 2–5 months, although exceptional patients without complete hematological remission have responses that last for 12–18 months. The improvement in platelet counts can be from pretreatment levels below 10 × 10^9^/L to stabilization at or above 20–50 × 10^9^/L.Responses can even be seen for patients with high-risk disease (e.g., relapsed AML) [22,37] and for patients with valproic acid serum levels below the therapeutic level.Toxic effects are usually dose-dependent and reversible, and the most common toxicities are gastrointestinal side effects and fatigue.

Thus, the valproic acid plus ATRA combination has clinically relevant antileukemic effects with acceptable toxicity in human AML, but the effect is usually limited unless low-toxicity chemotherapy is added.

## 5. Panobinostat—An Alternative HDAC Inhibitor to Valproic Acid in AML Therapy

Experimental studies suggest that the HDAC inhibitor panobinostat downregulates the expression of the transcription factor E2F1 and therefore suppresses the expressions of BRCA1, CHK1, and Rad51; this leads to an increased susceptibility to both cytarabine and daunorubicin-induced apoptosis [121]. Studies in AML cell lines suggest that panobinostat has largely overlapping and synergistic effects on gene expression profiles compared with DNA methyl transferase inhibitors [122]. Panobinostat has been tried in clinical studies in combination with standard AML induction therapy followed by panobinostat maintenance monotherapy [123]. The most frequent adverse events during monotherapy were thrombocytopenia and gastrointestinal side effects (nausea, vomiting, dyspepsia, and diarrhea), and similar side effects were observed when the drug was used after allogeneic stem cell transplantation for high-risk AML [124]. Two additional studies also suggest that panobinostat has moderate antileukemic activity, acceptable toxicity [125,126], and additional immunomodulatory effects with decreased number and activity of bone marrow infiltrating TNF receptor expressing Treg cells [127] but increased levels and function of other Treg cell subsets [128,129].

Only a few early clinical studies of panobinostat in AML have been published. First, it has been combined with intensive chemotherapy both for younger patients [125] and elderly patients receiving conventional intensive therapy [123,130]. It has also been combined with hypomethylating agents, including both decitabine [131], and azacitidine [132]. Third, it has been used as a maintenance treatment after intensive chemotherapy and could then reduce the effects of minimal residual disease for a subset of patients [123]. Finally, it has also been used as an additional posttransplant antileukemic treatment for patients receiving allogeneic stem cell transplantation for high-risk AML/MDS [124]. Taken together, these studies suggest that the addition of panobinostat can increase the toxicity, but the toxicity is still acceptable and this seems to be true also for allotransplant recipients.

## 6. The Treatment of Post-Allotransplant AML Relapse with ATRA, Valproic Acid, Hydroxyurea, and Azacitidine Illustrated by Four Case Reports

### 6.1. A Treatment Protocol for the Use of Valproic Acid and ATRA in Low-Toxicity Combination Therapy

To the best of our knowledge, there are no reports on the use of valproic acid + ATRA + azacitidine in allotransplant recipients. We therefore established a protocol where AML patients were included after written informed consent; the study was performed in accordance with the Helsinki declaration and approved by the regional ethics committee (REK Vest 2011/1257, REK 2011/1241) and The Norwegian Medicines Agency (Legemiddelverket; Ref. 11/09187-2 and code 343). Collection of the clinical data was registered by official Norwegian authorities (Personvernombudet, Haukeland University Hospital, registered 29092011), and the study was registered in public databases (ClinicalTrials.gov no. NCT01369368 and EudraCT no. 2011-002689-19). All patients had non-APL variants of AML diagnosed in accordance with the WHO criteria [133]. The diagnosis of AML relapse was based on the ELN criteria similar to previous studies [134].

This phase II study intended to include at least 10 patients. The scientific basis for the design of the study is described in Table 3. Our study included low-dose azacitidine and was stopped after a large retrospective study reported that monotherapy with higher azacitidine doses was effective and had acceptable toxicity when treating post-allotransplant AML relapse [135]. Our hypothesis was that sequential azacitidine/hydroxyurea combined with valproic acid + ATRA would be feasible as an outpatient treatment for unfit AML patients with leukemia relapse after allotransplantation. The treatment was given as five-week cycles until disease progression. First, valproic acid was started on day 1 and continued throughout the cycle; it was administered as an initial intravenous bolus dose of 5 mg/kg body weight and thereafter as an intravenous infusion for 24 h until oral treatment started; and the oral dose was increased until the therapeutic serum level or the maximal tolerated dose was reached. Second, oral ATRA 22.5 mg/m^2^ twice daily was given for the first 14 days of each cycle (day 1–14); this therapeutic strategy has previously been used for AML-stabilizing treatment [22,35,37,136]. Third, 5-azacitidine was administered as a subcutaneous injection at a fixed dose 100 mg per day on days 1–3 of each cycle [137]. Finally, hydroxyurea was administered from day 15 to day 35 of the first cycle; for later cycles, it was administered from day 5 to day 35. The hydroxyurea dose in the first cycle was 500 mg daily until day 21 and was thereafter increased to 1000 mg daily when no signs of response were observed [3,35,37,138]. Donor lymphocyte infusions were allowed but were not given to any patient.

### 6.2. What Should the Response Criteria Be for Patients Receiving AML Stabilizing Treatment?

There are no generally accepted response criteria for patients receiving AML-stabilizing treatment. In our opinion, one should use the generally accepted criteria for complete hematological remission, i.e., <5% blasts in the bone marrow together with normalization of peripheral blood cell counts [134]. In previous studies, we used peripheral blood cell counts to define disease stabilization [5,35,37]. First, stabilized platelet counts were defined as (i) an increase from lower platelet counts to stable values exceeding 15 or, (ii) for higher pretreatment levels, it was defined as an increase corresponding to a difference of at least 20 × 10^9^/L from the nadir level, with this difference being >20% of the nadir value. Second, increased neutrophil counts were defined as (i) an increase from <0.5 to >1.0 × 10^9^/L or, (ii) in the case of a higher pretreatment level, at least a doubling of the original value. Finally, increased reticulocytes were defined as (i) a doubling of the count or normalization of initially decreased levels. These effects had to be detected in at least two independent samples, and the duration of stabilization was defined as the time from first significantly altered levels until the first cell count below the pretreatment level.

### 6.3. Case Reports

Several studies have investigated the combination of hypomethylating agents and valproic acid in AML [19,113,138,139,140,141,142]. Most studies included elderly and unfit patients, and the most common valproic acid toxicities were encephalopathy and gastrointestinal side effects that seemed to be dose-dependent and reversible. However, many of these patients developed such side effects even at relatively low valproic acid serum levels. Only one of the studies observed an association between effect of treatment and high valproic acid serum levels [113], but response to valproic acid + ATRA treatment has been observed also for patients with serum levels below the therapeutic level (see Section 4). Important patient characteristics are summarized in Table 4.

#### 6.3.1. Case Report 1

The patient was a 22-year-old male with an early second AML relapse (complex karyotype) four months after re-transplantation for the first posttransplant relapse (Table 4). His ECOG performance status was 0 when he started antileukemic treatment according to the study protocol. He then had clinical signs of GVHD (skin and gastrointestinal tract); these symptoms increased when immunosuppression was reduced, and he therefore continued taking cyclosporine (target level 100–200 μg/L) and prednisolone at 12.5 mg daily. The ECOG status was 0 with minimal GVHD symptoms until the end of the third cycle. During the first cycle, he remained neutropenic and required erythrocyte transfusions, and his platelet count decreased to 21 × 10^9^/L. His platelet counts later increased to stable levels above 40–50 × 10^9^/L (maximal level 101 × 10^9^/L) and his peripheral blood blast counts were stable, but he still required regular erythrocyte transfusions. The median valproic acid serum level was 395 μmol/L (range 299–512 μmol/L) until he was admitted to a hospital by the end of the third cycle when he had signs of AML progression with musculoskeletal pains, increasing peripheral blood blast counts and severe thrombocytopenia. Progression occurred after 94 days of treatment, and he died 29 days later.

#### 6.3.2. Case Report 2

The patient was a 57-year-old female who was allotransplanted for AML in second complete remission with a matched unrelated donor. She developed acute GVHD in the skin early after transplantation but required only topical steroid treatment. AML relapse was diagnosed 71 days after transplantation (Table 4). Cyclosporine treatment was stopped, and she started antileukemic therapy according to the protocol. However, she developed increasing diarrhea during the first three weeks after inclusion and could no longer receive enteral nutrition or medication. Acute GVHD was suspected, and she started treatment with cyclosporine combined with systemic (prednisolone 30 mg/day) and topical oral steroids. However, her gastrointestinal symptoms continued and her ECOG status increased to 3–4. She died with a clinical picture of AML progression, severe acute GVHD, and pneumonia on day +100 posttransplant after 24 days of treatment for the relapse.

#### 6.3.3. Case Report 3

The patient was a 63-year-old male who developed secondary AML after treatment for non-Hodgkin’s lymphoma. He achieved complete hematological remission after the second induction cycle and was transplanted with reduced intensity conditioning (RIC). AML relapse was diagnosed less than four months posttransplant. Skin GVHD was diagnosed before the relapse and was treated with oral prednisone 30 mg daily in addition to topical steroids and continued cyclosporine therapy. Cyclosporine treatment was stopped when relapse was diagnosed, whereas steroid use was gradually reduced. The patient developed severe diarrhea three weeks after the start of the study treatment. Despite increased steroid treatment, the symptoms did not improve; he could not receive oral nutrition/medication and was removed from the study 46 days after inclusion. He died 21 days later due to AML progression, severe gastrointestinal GVHD, and pneumonia.

#### 6.3.4. Case Report 4

A 39-year-old female was diagnosed with atypical chronic myeloid leukemia (aCML). She received RIC conditioning and was transplanted from a HLA identical sibling. There was no acute GVHD, but she relapsed 20 months posttransplant with AML. At the start of treatment, she had 30% leukemic blasts in the bone marrow. She received azacitidine and hydroxyurea according to the protocol together with valproic acid (median serum level 432 μmol/L, range 334–496 μmol/L during the treatment period). Pretransplant peripheral blood cell counts showed total leukocytes 6.8 × 10^9^/L with mainly blast cells and severe thrombocytopenia <10 × 10^9^/L. Her platelets increased to 12 during the first cycle; at the same time, her leukocytes were 22.6 × 10^9^/L with neutrophils 5.9 × 10^9^/L. After the second azacitidine treatment, the dose of hydroxyurea was gradually increased to 1500 mg daily. The situation was stable with no signs of GVHD, total leukocytes in the range of 20–30 × 10^9^/L, normalized neutrophils 5.9 × 10^9^/L, but platelets <10 × 10^9^/L. She required erythrocyte transfusions every 2–3 weeks from inclusion until disease progression was diagnosed 98 days after start of the treatment. She died 30 days later.

#### 6.3.5. Comments to the Case Reports

All of our patients received a combination of sequential azacitidine/hydroxyurea, continuous valproic acid, and intermittent ATRA therapy. We emphasize that one should be very careful when making conclusions based on our case reports, but some suggestions can be made. First, especially case report 1 but also case report 4 suggest that our low-toxicity treatment stabilized the clinical status and could be administered while the patient was at home. Second, an acceptable quality of life with this outpatient treatment was then possible. Finally, the treatment probably did not induce GVHD, and this is in accordance with the previous studies of valproic acid effects on immunocompetent cells where immunosuppressive effects seem to dominate (see above). It is also in accordance with a previous pediatric study describing higher frequency of acute GVHD in patients with low vitamin A levels compared with higher levels [109]. However, for two of the patients with GVHD diagnosed before inclusion, a progression was seen after the start of treatment. We cannot exclude the possibility that ATRA and/or valproic acid contributed to this, although in our opinion, the reduction/end of ongoing immunosuppressive therapy when relapse was diagnosed was possibly more important. We also mention that a patient included in a previous study received valproic acid + ATRA + cytarabine and but did not develop GVHD either [138]. However, this question of GVHD development/progression has to be carefully addressed if valproic acid is further investigated in future clinical studies of posttransplant AML relapse.

To the best of our knowledge, no other data are available for such a combination treatment for posttransplant AML relapse, but a clinical study of low-dose treatment with the hypomethylating agent decitabine in combination with the alternative HDAC inhibitor panobinostat alone was recently published [131]. This treatment was used to intensify the antileukemic effects in patients without relapse, and these authors also concluded that the treatment was feasible with acceptable toxicity.

## 7. The Current Treatment for AML Relapse after Allogeneic Stem Cell Transplantation

Various strategies for posttransplant prophylaxis and treatment of AML relapse are summarized in Table 5 [2,143], and some of the most common strategies are described more in detail below. Less intensive/toxic strategies will often be tried in maintenance and pre-emptive therapy, whereas more intensive treatment can be used if the intention is remission induction followed by re-transplantation. Combined treatment can be used, for example, to stop GVHD prophylaxis combined with azacitidine and possibly donor lymphocyte infusion if the patient has no signs of GVHD [2,143,144].

### 7.1. The Hypomethylating Agents Azacitidine and Decitabine

As emphasized in a recent review [144], most clinical studies of hypomethylating agents in AML relapse after allotransplantation are relatively small and retrospective. The largest studies are summarized in Table 6 [135,137,144,145,146,147,148,149,150,151,152]. First, a response to this treatment is seen for a relatively large subset of patients, but these responses are usually of short duration and complete remissions are seen only for a relatively small fraction of patients. The maximal response can be reached as late as 100–150 days after the start of treatment, and remissions lasting for years have been observed for exceptional patients. Second, the most common and severe side effects are hematological toxicity, including neutropenic infections, but except for this, the treatment seems to be well tolerated. Third, the treatment has usually been combined with donor lymphocyte infusions (DLI), a treatment associated with a risk acute and chronic GVHD. Finally, azacitidine has been used in most studies, but decitabine is also effective and may induce responses even for patients who have failed on azacitidine. The doses differ between studies, and even relatively low doses of azacitidine can be effective (i.e., 100 mg fixed dose daily for three days). Thus, the use of hypomethylating agents and especially azacitidine should be regarded as a well-documented therapeutic alternative for patients with AML relapse after allogeneic stem cell transplantation. The possibility of combination therapy should also be further considered, as illustrated by a recent study combining lenalidomide with azacitidine [152].

### 7.2. Posttransplant Immunomodulation

The initial immunomodulatory strategy is to enhance the graft-versus-leukemia effect by reducing or ending immunosuppressive treatment. If significant GVHD or GVL effects do not occur after this intervention, DLI may be used to trigger GVL [144]. Several studies have used DLI in combination with azacitidine/decitabine (Table 6), and the risk of severe GVHD following this pharmaco-immunological intervention seems to be acceptable as long as DLI is not given too early posttransplant.

### 7.3. Intensive Induction Chemotherapy and Retransplantation

Intensive chemotherapy can be considered as a possible strategy especially for young patients that are fit enough to tolerate such treatments; if patients achieve complete remission, an allogeneic re-transplantation may be considered, but this decision should be individualized and should be based on careful evaluation of the risks of severe toxicity versus the possibility of maintained remission and cure [2,143]. This recommendation has also been given by other authors; patients with chemo-sensitive disease (i.e., achieving remission after initial chemotherapy) and having remission durations of at least 6–12 months after the first transplantation seem to be those who benefit the most, with long-term disease control after a second transplant [2,135,143,153].

## 8. Quality of Life for AML Patients

### 8.1. Quality of Life for AML Patients Receiving Intensive and Leukemia-Stabilizing Therapy

Patients receiving the best supportive care alone for AML spend 25–30% of their time in hospital and approximately 15% of their time attending clinic appointments [154]. Many of them have a high symptom burden [155], and one third of them are admitted to intensive care units during their initial symptoms [156]. Similar observations were also made in a third study; a significant proportion of elderly/unfit patients spend a significant part of their last days of life in hospital [157]. Thus, for patients without the possibility of being cured, the risk of additional toxicity causing a negative impact on the patients’ quality of life (QoL) by AML-directed chemotherapy may outweigh the possible advantage of an antileukemic effect.

Two studies suggest that elderly/unfit AML patients treated with hypomethylating agents show improved QoL compared with patients receiving only supportive care [158,159], but a major problem with one of these studies was a high rate of missing data [160]. The other study suggested that improvement in QoL was associated with response to treatment and delayed progression of the disease [159]. Studies of AML patients receiving more intensive treatment suggest that QoL declines during intensive treatment and thereafter recovers over time for patients achieving complete remission [160,161,162], but the impaired QoL may persist also after the end of the intensive treatment [163]. Allotransplant recipients who do not receive intensive treatment of their relapse will often have a shortened expected life time and a long-term reduction in their QoL due to the previous transplantation [164]; for such patients, it will be important to individualize the therapy and to carefully consider the treatment intensity and therefore the risk of toxicity from antileukemic therapy.

### 8.2. Quality of Life after Allogeneic Stem Cell Transplantation

Relatively few studies of health-related QoL are available for allotransplanted patients. One study investigated 115 patients during the first year posttransplant [165]; 72% of these patients had AML and more than 90% of them received peripheral blood stem cell grafts. The cohort included 51 patients receiving myeloablative conditioning (median age 41 years, range 21–60 years) and 64 patients receiving RIC conditioning (median age 59 years, range 43–69 years). The progression-free survival after 1 year (59 vs. 53%), overall survival, cumulative incidence of acute and chronic GVHD, relapse rate, and nonrelapse mortality did not differ between these two patient subsets. Both groups seemed to have a similar reduction in QoL when examined on day +30 posttransplant, and this was true both for the global QoL score as well as physical, role, social, cognitive, and emotional functioning. There was a slow recovery back to baseline QoL pretransplant levels during the first year posttransplant without any differences between the two groups. Furthermore, QoL one year posttransplant was also investigated in the MRC-AML10 study [166]. In this study, allotransplantation seemed to have an adverse impact on most QoL parameters compared with intensive consolidation chemotherapy or autotransplantation. Finally, another larger study also reported reduced QoL for allotransplant recipients compared with patients who received intensive chemotherapy, and this was especially true for patients receiving continued immunosuppressive treatment for GVHD [167]. Taken together, these studies suggest that many allotransplanted patients already had reduced QoL when relapse was diagnosed, and this has to be considered when deciding the therapeutic strategy for relapse. This is especially important for patients with aggressive early relapse and shortened expected survival. In this context, it should also be emphasized that valproic acid/ATRA may have antileukemic effects even when the maximal tolerated dose is low and the recommended therapeutic serum level is not reached [3,35,37,114,115].

## 9. Summarizing Comments

### 9.1. Alternative Pharmacological Combinations

Leukemia relapse is a common cause of death in allotransplanted AML patients. The risk of relapse may possibly be reduced by additional posttransplant chemotherapy, e.g., kinase inhibitors [168], low-dose or late lenalidomide [169], low-dose azacitidine [170,171] or its oral formulation CC-486 [172], decitabine [173], gemtuzumab [174], or prophylactic DLI in patients without clinical GVHD [175] (see the previous Table 5). Examination of posttransplant minimal residual disease may become useful to guide such therapeutic strategies [176,177]. Although these approached may reduce the risk of relapse, the treatment of posttransplant AML relapse will still be a challenge. Many of these patients are not fit for intensive treatment, and AML-stabilizing treatment will then be their only alternative.

The efficiency of hypomethylating agents and especially azacitidine as an AML-stabilizing treatment for posttransplant AML relapse was documented in several clinical studies (Table 7) [135,137,144,145,146,147,148,149,150,151,152]. Even low-dose therapy can be effective, and the toxicity seems acceptable. Furthermore, ATRA has been incorporated in several clinical AML trials but not for allotransplant recipients and with divergence; taken together, these results suggest that ATRA therapy has low toxicity and may contribute to the antileukemic effect of combined treatment at least for certain patients or patient subsets [4]. The combination of valproic acid and ATRA has antileukemic activity and low toxicity [35,37]. Experimental studies also suggest that valproic acid (possibly in combination with ATRA) can be combined with other antileukemic agents (Table 7) [122,178,179,180,181,182,183,184,185]. The addition of these two agents to conventional chemotherapy (e.g., azacitidine and even a fourth antileukemic agent, for example venetoclax) should, in our opinion, be further explored in future clinical trials, and this is supported by our present case reports.

### 9.2. Alternative HDAC Inhibitors

Valproic acid as well as several other HDAC inhibitors are now considered for the treatment of AML, but even the newer inhibitors seem to have a limited antileukemic effect when used as monotherapy [186,187,188]. These new HDAC inhibitors should be further considered as possible alternatives to valproic acid for the treatment of posttransplant relapse, but one should emphasize that, for several of them, the toxicity has to be carefully considered because the side effects seem more severe than that for valproic acid [189,190]. However, for belinostat, the toxicity seems to be acceptable, at least for T cell lymphoma patients where fatigue and gastrointestinal toxicity are most frequent and the risk of severe hematological toxicity is low [191]. Finally, valproic acid modulates systemic metabolism, and such metabolic effects may contribute to antileukemic effects [5] and be unique for valproic acid (a fatty acid) compared to other HDAC inhibitors.

### 9.3. Possible Effects on Graft Versus Leukemia (GVL) Effects

As described in Section 2 and Section 3, both ATRA and valproic acid have effects on several immunocompetent cells, and as outlined above, these drugs may also influence the risk of GVHD. The basis for the use of donor lymphocyte infusions in several of the clinical studies (see Section 7.1 and Table 6) is to increase antileukemic GVL activity without the development of severe GVHD [2,134,143]. It would not be surprising if the immunomodulatory effects of ATRA and/or valproic acid could influence posttransplant GVL activity, but to the best of our knowledge, this question has not been addressed in previous studies and it is difficult to provide an answer based on the available scientific literature.

Checkpoint inhibitors are now being tried in the treatment of AML [192]. This therapeutic strategy is also considered for posttransplant enhancement of antileukemic GVL activity, but the conclusion from small clinical studies is that this treatment can exacerbate GVHD [193]. The recently published experience for 209 patients with Hodgkin disease allotransplanted after PD-1 blockade may then be relevant [194]. The 180-day cumulative incidence of grade 3–4 acute GVHD was 15% and the 2-year incidence of chronic GVHD was 34%. Furthermore, a longer interval from PD-1 blockade to allotransplantation was associated with decreased risk of severe acute GVHD, whereas additional treatment between PD-1 blockade and transplantation was associated with increased relapse risk. Finally, the use of cyclophosphamide-based GVHD prophylaxis was associated with GVHD-free and relapse-free survival. These observations suggest that there is a risk of severe immune-mediated toxicity when combining checkpoint inhibition and allotransplantation, but optimal timing and transplantation procedures seem to make combined treatment possible and therefore enhance posttransplant anticancer reactivity. However, it is not known whether it will be possible to use this strategy in AML and whether/how ATRA or valproic acid will interfere with such a GvL targeting therapy.

## 10. Conclusions

To answer our initial question, in our opinion, valproic acid and ATRA should be further investigated for the treatment of post-allotransplant AML relapse. It is a unique combination with antileukemic effects that may be mediated both through systemic metabolic modulation and through direct effects on the leukemic cells. However, neither of the two drugs should be used as monotherapy but should be used as a part of a low-toxicity AML-stabilizing treatment, especially for unfit patients, e.g., patients with reduced posttransplant quality of life. It should also be emphasized that reducing the risk of relapse through posttransplant antileukemic pharmacotherapy and/or DLI needs to be further explored [130,143,195] together with alternative epigenetic strategies [196] and epigenetic approaches based on venetoclax [197].

## Figures and Tables

**Figure 1 pharmaceuticals-14-00423-f001:**
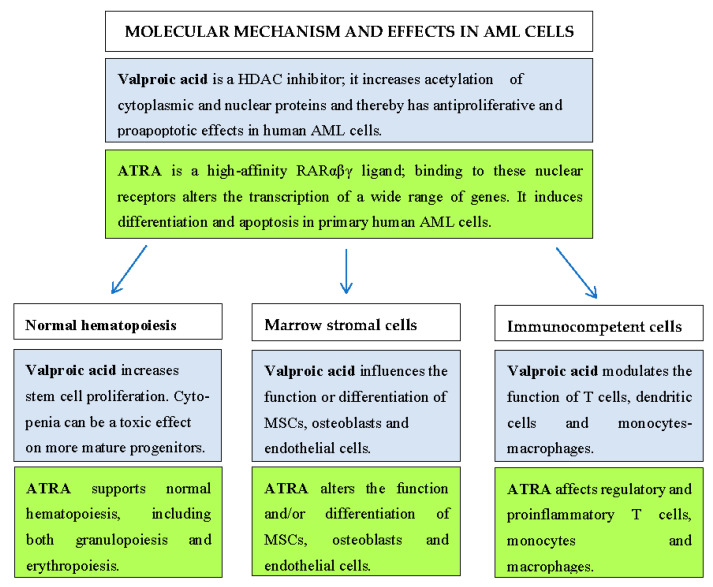
A summary of important effects of valproic acid and ATRA on AML cells, normal hematopoiesis, bone marrow stromal cells, and immunocompetent cells. For a detailed and more complete description of the pharmacological effects including references, please see Section 2 and Section 3.

**Table 1 pharmaceuticals-14-00423-t001:** Important characteristics of valproic acid and all-trans retinoic acid (ATRA) [3,4,5,6,7,8,9,10,11,12,13,14,15,16,17,18].

	Valproic Acid	ATRA
Chemical classification	Branched short-chain fatty acid	Vitamin A derivative/retinoid
Molecular target	Regulates transcription through inhibition of histone deacetylation and therefore increased gene transcription. HDACs are grouped into class I (HDACs 1/2/3/8), class II (HDACs 4/5/6/7/9/10), class III (the sirtuins SIRT1-7), and class IV (only HDAC11). Valproic acid inhibits class I and class II HDACs, but HDACs 9/11 are activated and HDACs 6/8/10 are not affected.	The retinoic acid receptors (RARs) and retinoid X receptors (RXRs) are ligand-activated nuclear receptors. ATRA is a high-affinity activating ligand of the three RARα/β/γ receptors and is involved in the transcriptional regulation of hundreds of genes.
Cellular effects	Altered protein acetylation, including altered histone acetylation and altered acetylation of many other cellular proteins including cytoplasmic proteins	Altered transcriptional regulation mediated by its binding to the nuclear RAR receptors
Possible interactions at the cellular level	Valproic acid seems to influence the differentiation-inducing effect of ATRA; it also increases aldehyde dehydrogenase activity and may therefore have effects on vitamin A metabolism.	There is a molecular crosstalk between HDACs/histone acetylases and ATRA; ATRA also influences gene expression through epigenetic modulation.
Effects on normal hematopoiesis	Valproic acid seems to stimulate stem cell proliferation. Experimental studies suggest that it upregulates genes important for myelomonocytic differentiation but inhibits expression of genes important for erythroid differentiation. Drug-induced cytopenias are usually dose-dependent and reversible.	A majority of studies suggest that ATRA promotes the activity of hematopoietic stem cells and regulates differentiation of hematopoietic progenitor cells.
Systemic metabolic effect	Altered serum levels of several amino acid and fatty acid metabolites	Altered serum levels of several amino acid and fatty acid metabolites
Administration	Oral or intravenous	Oral
Monitoring of doses	Defined therapeutic serum levels make monitoring possible	Dosing based on body surface area
Accepted indication	Epilepsy, depression	Treatment of APL
Previous clinical studies in non-APL variants of AML	Mainly phase I/II studies of AML-stabilizing treatment, often in combination with ATRA	Phase I/II clinical studies of AML stabilizing treatment and randomized clinical studies in combination with intensive chemotherapy

Abbreviations: APL, acute promyelocytic leukemia; HDAC, histone deacetylase.

**Table 2 pharmaceuticals-14-00423-t002:** Proteomic studies of patients receiving anti-AML therapy based on ATRA plus valproic acid. Proteomic and phosphoproteomic comparisons of AML cells derived from responders and nonresponders and leukemic cells derived before and during ATRA/valproic acid therapy [32].

Pretreatment Differences between Responders and vs. Nonresponders	
Proteomic Effects	Phosphoproteomic Effects
*Responders:* High levels of proteins reflecting neutrophil differentiation, intracellular transport, p53 signaling, and amino acid metabolism *Nonresponders:* high levels of proteins involved in transcription, cytoplasmic organelles, and lipid metabolism	*Responders:* High phosphorylation of proteins reflecting cytoskeleton organization, cell cycle regulation, membrane bound, and extracellular organelles *Nonresponders:* high phosphorylation of proteins involved in RNA binding/splicing
**in vivo effects of ATRA**	
*Responders:* Altered ribonucleoprotein assembly, RNA biosynthetic process, histone modification, chromatin binding, kinase activity, and signal transduction *Nonresponders:* altered extracellular secretion/organelles, clathrin complex, cytoskeleton organization, and RNA function/synthesis	*Responders:* Altered RNA processing and splicing, Rho GTPase binding, Rac GTPase binding, transcription elongation, and ribosome *Nonresponders:* RNA splicing and processing, nuclear body, and plasma membrane
**Further modulation by adding valproic acid to ATRA in vivo**	
*Responders:* Further modulation of mitochondrial function, ribonucleoprotein assembly, and kinase activity *Nonresponders:* RNA binding and splicing, mRNA function, and clathrin complex	*Responders:* Altered RNA processing, nuclear speck, and actin filament binding; further modulation of Rac GTPase binding and transcription elongation *Nonresponders:* Altered cyclin-dependent protein ser/thr kinase activity, and cellular catabolic process; further modulation of RNA splicing/processing

**Table 3 pharmaceuticals-14-00423-t003:** A summary of the scientific basis for the design of the present protocol for treatment of posttransplant AML relapse based on ATRA plus valproic acid.

**Valproic acid** [35,37,114,115] Treatment started on day 1 of the first cycle and continued until disease progression. Early start of valproic acid therapy was regarded as important because it will often take 2–3 weeks before a response can be detected. First administered as intravenous infusion to reach a relevant therapeutic level as early as possible. The dose was increased until one reached the highest level with acceptable toxicity or the recommended therapeutic serum level. Previous studies suggest that systemic levels lower than the therapeutic serum level can be effective.
**ATRA** [35,37] Administered for days 1–14 of each five-week cycle. The daily dose was 22.5 mg/m^2^ twice daily; this is similar to the dose used in APL. A fourteen day duration has been used in previous AML studies of ATRA/valproic acid therapy. ATRA syndrome has not been observed with this dose and duration of treatment.
**Azacitidine** [137] Included patients were regarded as unfit for more intensive treatment. The risk of severe toxicity was reduced using a fixed dose of 100 mg daily for the first three days of each cycle. The three-day low-toxicity regimen has a clinically relevant antileukemic effect in posttransplant AML relapse and can be effective even in patients with high-risk karyotypes. The interval (i.e., duration of each cycle) was increased to five weeks to further reduce the risk of severe toxicity. Azacitidine started early from day 1 because it may last several weeks before a response is seen (see Section 7.1).
**Hydroxyurea** [35,37,138] The use of hydroxyurea in addition to ATRA/valproic acid was allowed in previous studies. It started on day 15 of the first cycle to reduce the risk of initial severe toxicity. In later cycles, it was administered at days 5–35 to avoid overlap with the azacitidine. To reduce the risk of severe hematological toxicity, hydroxyurea dosing was guided by normal peripheral blood cell counts and the level of circulating AML blasts in peripheral blood. If this leukocytosis could not be controlled (defined as >50 × 10^9^/L), we changed it to oral merkaptopurin or subcutaneous cytarabine, and we then used the same guideline as for hydroxyurea for dosing of these two drugs [138].

**Table 4 pharmaceuticals-14-00423-t004:** Important characteristics for patients receiving stabilizing treatment for relapse AML after allotransplantation; the status at inclusion in the study.

	CASE 1	CASE 2	CASE 3	CASE 4
Age/gender	22 years/male	57 years/female	63 years/male	55 years, female
Status at inclusion	Second relapse 122 days after second allo-SCT	Second relapse 71 days after allo-SCT	First relapse 113 days after allo-SCT	AML relapse 20 months after allo-SCT for aCML
Status at allo-SCT	Second remission	Second remission	First remission	Remission
Stem cell donor	HLA identical sibling donor	MUD	HLA identical sibling	HLA identical sibling
Conditioning	Myeloablative	Myeloablative	Reduced intensity	Reduced intensity
Bone marrow blasts	56%	30%	>20% on biopsy	21%
Acute GVHD	Skin, GI-tract	Skin, GI tract	Skin, GI tract	No
ECOG status	0	2	0	0
**Pretreatment peripheral blood cells/transfusions**				
Hemoglobin	12.0 g/100 mL	10.5 g/100 mL	10.2 g/100 mL	10.1 g/100 mL
Neutrophils (×10^9^/L)	2.4 × 10^9^/L	4.5 × 10^9^/L	1.0 × 10^9^/L	0.5 × 10^9^/L
Platelets (×10^9^/L)	88 (decreasing to 21) × 10^9^/L	13 × 10^9^/L	6 × 10^9^/L	12 × 10^9^/L
AML blasts (×10^9^/L)	2.5 × 10^9^/L	1.5 × 10^9^/L	<1% of leukocytes	<0.2 × 10^9^/L
Valproic acid levels during first cycle		253 μmol/L after iv and 40–50 μmol/L after oral administration	Exceeding 400 μmol/L after iv but <100 μmol/L after oral administration	
Red cell/platelet transfusions first cycle	2/0	Cycle 1 not completed	Cycle 1 not completed	
**Survival**				
From (last) allo-SCT	240 days	100 days	132 days	23 months
From relapse	133 days	29 days	62 days	110 days
From start of therapy	128 days	24 days	46 days	128 days
From progression	29 days	No response	No response	30 days

Abbreviations: aCML, atypical chronic myeloid leukemia, GI, gastrointestinal; iv, intravenous; LDH, lactate dehydrogenase; MUD, matched unrelated donor; SCT, stem cell transplantation.

**Table 5 pharmaceuticals-14-00423-t005:** Prophylaxis and treatment of post-allotransplant AML relapse. The table presents the definition of relapse, the importance of the clinical status at the time of intervention, and the various pharmacological and immunological strategies for prophylaxis and treatment [2,134,143,144].

**Morphological definition of AML relapse**
At least 5% myeloblasts in the bone marrow in a patient with previous diagnosis of AML; if less than 10% of blasts, the increased blast count should be verified in a second bone marrow sample.
**Therapeutic strategy depending on the status at the time of intervention**
Maintenance therapy: No evidence for residual disease, i.e., molecular remission. Pre-emptive therapy: Detection of posttransplant minimal residual disease. Salvage therapy: Morphological signs of relapse.
**Pharmacological strategies**
Intensive chemotherapy, e.g., conventional induction chemotherapy that results in complete remission usually of short duration for 25–30% of patients. Low-toxicity chemotherapy, e.g., hypomethylating agents that results in 15–20% complete remissions. Targeted therapy: The pharmacological agent depends on the AML-associated genetic abnormalities, e.g., IDH1/IDH2 inhibitors when IDH mutations and Flt3 inhibitors in patients with Flt3-ITD. The bcl2-inhibitor venetoclax is now investigated in clinical trials. The HDAC inhibitor panobinostat is in clinical trials.
**Antileukemic immunomodulation**
Early reduction of GVHD prophylaxis Donor lymphocytes infusion Re-transplantation after remission induction

**Table 6 pharmaceuticals-14-00423-t006:** Important studies investigating hypomethylating agents for treatment of relapsed AML after allotransplantation.

Study	Treatment	Effect	Toxicity and GVHD
Lübbert et al. [137]. Retrospective (*n* = 26). Time from transplant to relapse median 248 days (55–1412 days).	Azacitidine 100 mg total daily dose, days 1–3 with 21 days intervals, could be followed by DLI was allowed. Median number of cycles 2 (range 1–10).	Complete remission in 4 patients with duration 450–820 days. 50% with temporary disease control. Median survival 136 days	Neutropenic infections in 4 patients. Acute GVHD 2 patients.
Czibere et al. [145]. Retrospective (*n* = 22). Time from transplant to relapse|03 days (53–708 days).	Azacitidine 100 mg/m^2^ daily days 1–5, 2–5 weeks interval, DLI was allowed. Median number of cycles 2 (range 1–8).	Response to azacitidine: 16 patients. Median time to relapse 433 days (range 114–769 days); median survival 144 days.	Acute GVHD in 6 patients. Hematological toxicity grade 4 in 7 patient; neutropenic infections in 6 patients.
Steinmann et al. [146]. Retrospective (*n* = 72). Time from allograft to relapse 253 days (50–2126 days).	Azacitidine 100 mg daily for 3 days (5 days in first cycle if leukocytosis) and repeated every 3 weeks. Median number of cycles 2.7; 65 patients received DLI	Complete remission 9.7% (for two patients lasting >5 years), temporary disease control 44%. Median survival 108 days; peripheral blood blasts <1% predicted longer survival.	10 patients developed acute GVHD; 9 patients with chronic GVHD. 2 patients with therapy-related sepsis. 28/72 hospitalized due to infections; 15 patients with grade 3/4 neutropenia.
Tessoulin et al. [147]. Retrospective (*n* = 31). Time from transplant to relapse median 3.7 months (range 1.7–37.6 months).	Azacitidine 75 mg/m^2^ daily for 7 days every 4 weeks. DLI was allowed. Median number of cycles 3 (range 1–12).	4 complete remissions. 11/31 patients responded to the treatment. Median overall survival 153 days.	Grade 3/4 toxicities in 36% of patients. 38% readmitted to hospital, the most common reasons being fever, infection, or neutropenia.
Schroeder et al. [148]. Retrospective (*n* = 154). Time from transplant to relapse median 185 days (19–3349 days).	Azacitidine 50–100 mg/m^2^ for 5–7 days every 4 weeks. DLI allowed. Median number of cycles 4 (4–14).	Complete remission 27%; overall response rate 33%. Better overall survival for patients with MDS or <13% bone marrow blasts; 2-years survival 29%.	Acute GVHD 23%. Chronic GVHD 27%.
Craddock et al. [135]. Retrospective (*n* = 181). Time from transplant to relapse median 8 months (1–71 months).	Azacitidine 75 mg/m^2^ for 5–7 days every 4 weeks; DLI was allowed. Median duration of treatment 53 days (range 2–1196 days).	29.3% with complete or partial remission. Higher response rates for patients transplanted in remission and for MDS. Longer survival for patients with low blast counts (<20%) or >6 month from transplant to relapse.	
Woo et al. [149]. Prospective (*n* = 39). Patients with morphological relapse or evidence for persisting disease (flow cytometry, karyotyping) before day +100.	Azacitidine 75 mg/m^2^ daily for 7 days every 4 weeks (≥6 cycles; DLI allowed). Median bone marrow blasts 1.4%.	3 patients with complete remission; 9 patients with partial remission. 3 additional patients with stabilization. 2-years overall survival 25%.	Acute GVHD 29 patients, 4 with grade III.
Sommer et al. [150]. Retrospective (*n* = 26). Time from transplant to relapse median 306 days (76–4943).	Decitabine 20 mg/m^2^ daily for 5 or 10 days every 4 weeks; one DLI during each cycle was allowed. Median number of cycles 2 (range 1–13).	Complete remission 4/26, partial remission 1/26. Median overall survival 4.7 months.	Acute GVHD 17%. Chronic GVHD 6%.
Schroeder et al. [151]. Retrospective (*n* = 36). Median time to relapse 370 days (43 2623).	Decitabine 20 mg/m^2^ daily for 5 or 10 days with 4 weeks interval.; DLI was allowed Median number of cycles 2 (range 1 11).	Overall response rate 25% including 6 patients with complete remission. Two patients who failed azacitidine reached complete remission.	Acute GVHD in 7 patients; for 2 of them no previous DLI. 2 patients with symptoms of chronic GVHD. after DLI.
Craddock et al. [152]. Prospective (*n* = 29). Median time to relapse 10 months (1–39 months).	Azacitidine 75 mg/m^2^ days 1–7 followed by lenalidomide (25 mg daily determined as the maximal tolerated dose) days 10–30. Median number of cycles 3 (0–11).	Among 15 patients receiving at least 3 cycles, there were 3 complete remissions, 3 with complete remission without complete regeneration, and 1 partial remission. Median overall survival was 27 months for responders and 10 months for nonresponders.	3 patients developed acute GVHD grade 2–4. Febrile neutropenia observed in 10 patients; documented infections in 3 patients and sepsis in 7 patients.

**Table 7 pharmaceuticals-14-00423-t007:** Combination of valproic acid with other agents: a summary of experimental studies [122,178,179,180,181,182,183,184,185].

Study	Study Design	Observations
Liu et al. [178] Cytarabine	In vitro studies of AML cell lines and primary human AML cell lines	Increased *BAX* expression detected at the mRNA level leading to reduced AML cell proliferation, sub-G1 arrest, and apoptosis
Xie et al. [179] Cytarabine	In vitro studies of pediatric t(8;21) positive and negative AML cells	t(8;21)-positive cells were most susceptible to the combined treatment with induction of DNA double-stranded breaks together with induction of Bim-mediated and caspase-dependent apoptosis.
Leitch et al. [180] Hydroxyurea, an antimetabolite causing stalling of S-phase replication forks	AML cell lines and primary patient cells evaluated in vitro and in xenograft models	Valproic acid amplified the ability of hydroxyurea to slow S-phase progression; this effect was correlated with increased DNA damage. Reduced expression of the DNA repair protein Rad51
Blagitko-Dorfs et al. [122] DNA methyl transferase inhibitors	AML cell lines	Combined treatment affected more transcripts than the sum of the genes altered by either treatment alone; downregulation of oncogenes and epigenetic modifiers
Xie et al. [185] Clofarabine, a second generation purine nucleoside analog	Pediatric cell lines and primary samples studied in vitro	Synergistic antileukemic effects in cells sensitive to valproic acid with Bax activation and apoptosis; for valproic acid-resistant cells, antagonistic effects were observed.
Heo et al. [181] Dasatinib, a multi-targeted kinase inhibitor	AML cell lines and patient samples investigated in vitro	Synergistic effects with G1 phase cell cycle arrest and induction of caspase-dependent apoptosis. The effects seemed to be mediated by MEK/ERK and p38 MAP kinases.
McCormack et al. [182] Nutlin 3, a MDM2 antagonist	AML cell lines and patient cells tested in vitro and *in xenograft* models	Cotreatment resulted in the induction of p53, acetylated p53, and p53 target genes compared with either agent tested alone; this was followed by p53-dependent cell death with autophagic features.
Wang et al. [184] Bortezomib, a proteasome inhibitor	In vitro studies of AML cell lines and primary AML cells	Antiproliferative and proapoptotic effects with increased mitochondrial injury and caspase activation. They observed reduced NFκB, Akt, and ERK signaling and activation of stress-induced pathways (e.g., JNK and p38). The combination caused G2/M arrest and increased p21.
Nie et al. [183] Bortezomib	The HL60 cell line	G0/G1 arrest with induction of apoptosis, inhibition of cyclin D1, and telomerase activity

## Data Availability

The data presented in this study are available upon request from the corresponding author.

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
