# Peer review of "Therapeutic Use of Valproic Acid and All-Trans Retinoic Acid in Acute Myeloid Leukemia—Literature Review and Discussion of Possible Use in Relapse after Allogeneic Stem Cell Transplantation"

_pharmaceuticals, 2021, doi:10.3390/ph14050423_

Round 1

Reviewer 1 Report

Comments to the authors:

This manuscript provides an overview of the general experience with valproic acid/ATRA in AML therapy and discusses its potential use in low-intensity/toxicity treatment of post-allotransplant AML relapse. The authors discussion is further illustrated by four case reports where combined treatment with sequential azacitidine/hydroxyurea, valproic acid and ATRA was used. Overall, the topics presented in this manuscript are informative and well-written; however, for better understanding of this topic to a broad readership, I suggest that the authors provide either a table or simplified figure to represent the study findings briefly. A number of minor concerns have been noted, which should be addressed prior to the acceptance for publication. Some suggestions are provided below:

  1. The authors should briefly describe the chemical structures, and pharmacological and physiological mechanisms of Valproic acid, ATRA, and their metabolites, along with a simplified illustrative diagram.
  2. For a better understanding of this topic to the readers, the authors should provide a table summarizing and listing the experimental, pharmacological mechanism, clinical studies, and some details of valproic acid and ATRA in non-APL variants of AML (include effect on AML cells, normal hematopoietic and bone marrow stromal cells, and normal immunocompetent cells). In addition, the combined treatment should also be considered in it.
  3. Finally, please provide a table listing or a simple diagram illustrating the overview about the current AML therapy and relapse treatment in patients.

Author Response

This manuscript provides an overview of the general experience with valproic acid/ATRA in AML therapy and discusses its potential use in low-intensity/toxicity treatment of post-allotransplant AML relapse. The authors discussion is further illustrated by four case reports where combined treatment with sequential azacitidine/hydroxyurea, valproic acid and ATRA was used. Overall, the topics presented in this manuscript are informative and well-written; however, for better understanding of this topic to a broad readership, I suggest that the authors provide either a table or simplified figure to represent the study findings briefly. A number of minor concerns have been noted, which should be addressed prior to the acceptance for publication. Some suggestions are provided below:

1. The authors should briefly describe the chemical structures, and pharmacological and physiological mechanisms of Valproic acid, ATRA, and their metabolites, along with a simplified illustrative diagram.

Response: A new Table 1 is included with more detailed general information about the two drugs; 14 new references are included and are given in the heading of this table. A brief comment is added in the text; the new references are also included here (lines 60-65).

2. For a better understanding of this topic to the readers, the authors should provide a table summarizing and listing the experimental, pharmacological mechanism, clinical studies, and some details of valproic acid and ATRA in non-APL variants of AML (include effect on AML cells, normal hematopoietic and bone marrow stromal cells, and normal immunocompetent cells). In addition, the combined treatment should also be considered in it.

Response: It would be difficult to include all this information in a single Table/figure. However, to summarize and illustrate the complexity of the pharmacological effects for these two drugs we have included a new Figure 1. Due to the more detailed information required to make this figure complete, we added a brief statement and ne references (lines 297-299). The figure is referred to and commented in the new section 3.5.

The clinical observations were not included in Figure 1. To help the reader we have instead restructured section 4 with bullet points to make this easier for the reader. We hope this solution is acceptable.

As a response to the final comment on combined treatment we have included a new Table 3 where we describe more in detail the scientific basis for the design of our own combined treatment; in our opinion this basis is of general interest and should also be considered in the design of future clinical studies. We agree with the reviewer that this should be better addressed, but we hope that our solution with a separate table in addition to Figure 1 can be accepted. The new table is referred to in the text (lines 478, 479).

We have also included a new section and Figure 1

3. Finally, please provide a table listing or a simple diagram illustrating the overview about the current AML therapy and relapse treatment in patients.

Response: We have included a new Table 5 where we describe briefly the general principles for posttransplant prophylaxis and treatment of AML relapse. We have also included a brief new chapter in section 7 (lines 612-615). The table is also referred to in line 722.

Reviewer 2 Report

Overall, this is a well-written and fairly comprehensive review on the use of valproic acid and ATRA as treatment for AML. Specifically, this potential treatment in the context of relapse and transplantation with low-intensity treatment is discussed and illustrated with case studies. Significant discussion on the immunological effects of this therapy as related to GvHD is included; however, the related issue of GvL is barely commented on. The authors should expand upon the dichotomous effects of GvHD/GvL and how this proposed treatment might impact (positively or negatively) patients receiving VPA/ATRA therapy. 

Minor points:

  • The title should be more succinct
  • This review contains nice summary tables. It would also be helpful if the authors had at least 1 figure that might illustrate the overall concept (or a major aspect of) this review.

Author Response

Overall, this is a well-written and fairly comprehensive review on the use of valproic acid and ATRA as treatment for AML. Specifically, this potential treatment in the context of relapse and transplantation with low-intensity treatment is discussed and illustrated with case studies. Significant discussion on the immunological effects of this therapy as related to GvHD is included; however, the related issue of GvL is barely commented on. The authors should expand upon the dichotomous effects of GvHD/GvL and how this proposed treatment might impact (positively or negatively) patients receiving VPA/ATRA therapy. 

Response: We agree that the question of GVL is important, but to the best of our knowledge this question has not been addressed in previous studies. It is also very difficult to discuss this in detail without becoming speculative. However, we have included a brief comment in a separate section to make this clear (lines ). The summarizing section has been restructured without altering other parts of the text to make this GvL comment visible.

Minor points:

1. The title should be more succinct

The original title had 226 signs, the present has 191 signs

2. This review contains nice summary tables. It would also be helpful if the authors had at least 1 figure that might illustrate the overall concept (or a major aspect of) this review.

The following changes have been made:

  • A new Table 1 is included.
  • Important observation with regard to AML cells and normal cells are summarized in a new Figure 1.
  • The scientific basis for our patient treatment is listed in a new Table 3.
  • As required by the other reviewer general strategies for prophylaxis and treatment of posttransplant relapse is described in a new Table 5.